# Bibliometric Analysis of the Toxicity of Bisphenol A

**DOI:** 10.3390/ijerph19137886

**Published:** 2022-06-27

**Authors:** Mengmei Ni, Xiaomeng Li, Lishi Zhang, Vikas Kumar, Jinyao Chen

**Affiliations:** 1West China School of Public Health, West China Fourth Hospital, and Healthy Food Evaluation Research Center, Sichuan University, Chengdu 610041, China; 2019324040011@stu.scu.edu.cn (M.N.); lixm516@scu.edu.cn (X.L.); lishizhang@scu.edu.cn (L.Z.); 2Environmental Engineering Laboratory, Departament d’ Enginyeria Quimica, Universitat Rovira i Virgili, Av. Països Catalans 26, 43007 Tarragona, Spain; 3Institut d’Investigació Sanitària Pere Virgili (IISPV), Hospital Universitari Sant Joan de Reus, Universitat Rovira i Virgili, 43201 Reus, Spain

**Keywords:** BPA, toxicity, bibliometric analysis, VOSviewer

## Abstract

Bisphenol A (BPA) is used worldwide and research on the toxicity of BPA has advanced rapidly in the last few decades. This study aimed to evaluate the global scientific output of toxicity of BPA and explore the hot spots and research trends. All available articles related to the toxicity of BPA until 2022 were retrieved from the Web of Science Core Collection database. The VOSviewer, a bibliometric analysis software, was used to analyze the information of included articles, including countries/institutions, international cooperation, journals, citations, and keywords. Among 1644 retrieved articles, 1611 eligible studies were identified for analysis, and the annual publications increased with time in the past three decades. China and the United States were the most active contributors in this field. Chinese Academy of Sciences and the Dow chemical company conducted relatively more research than others about BPA toxicity. The journal “*Chemosphere*” published the most studies on “BPA toxicity”. Before 2015, most research focused on estrogenic activity and the test system mainly utilized animal experiments. However, in recent years, research related to toxic mechanisms of BPA at the cellular level and the toxicity of its analogs have received widespread attention. Considering some critical research gaps, future research on BPA toxicology should probably focus on the molecular biology of toxic mechanism, mixture toxicity, and co-exposure of BPA substitutes. This study will help researchers understand past and current research trends, hot spots, and trends of toxicity studies of BPA and, thus, contribute to further research and risk management of BPA.

## 1. Introduction

Bisphenol A (BPA) is widely used as a monomer or additive in producing polycarbonate plastics, epoxy resins, other polymeric materials, and paper products [1,2]. It has been found in the environment (sediment, sludge, water, air, dust), foods (beverages, meat, cereals, vegetables, etc.), consumer products (paper products and personal care products), and human biological samples [3]. People can be exposed to BPA through oral means, skin, and inhalation [4]. BPA is a typical environment/food contaminant and a classic endocrine disruptor [5]. Studies have shown that BPA exerts many kinds of harmful effects [6], including reproductive and developmental toxicity, neurotoxicity [7], and immunotoxicity [8,9], and may even increase the risk of some cancers [10,11].

Bibliometric analysis, a method in literature analysis, can be used to analyze and visualize a large amount of literature information, including author, title, keywords, institutions, citation information, publishing date, collaborator, publisher, and so on [12]. This method is more efficient and intuitive than traditional literature analysis, which only qualitatively summarizes literature by researchers themselves [13]. Some popular literature analysis software currently available are VOSviewer, Citespace, CitNetExplorer, etc. Among them, VOSviewer has been used successfully in various projects carried out by the Centre for Science and Technology Studies [14] and different research fields, such as “COVID-19” [15], agricultural pollution [16], neuroarchitecture [17], and so on [18,19]. As a freely available computer program for constructing and viewing bibliometric maps, VOSviewer could analyze the results in an easy-to-interpret way [14,20,21]. Su et al. [22] used VOSviewer to evaluate nanoparticles’ global scientific output of neurotoxicity and explore their hot spots and research trends. Zuanazzi et al. [23] summarized the available data to provide insights into 2,4-D toxicity and mutagenicity characteristics. de Castilhos et al. [24] analyzed the publication patterns of main topics related to Gly research with scientometrics tools and found “toxicology” was the most influential area.

However, there is no study reporting BPA toxicity based on bibliometric analysis. The research on BPA toxicity has advanced rapidly. Before delving into BPA toxicity, it is crucial to preliminarily analyze the published papers, map the research profile, and explore hot spots and trends in this field. Therefore, this study used VOSviewer to analyze the literature about BPA toxicity in order to depict the research profile of BPA toxicity and explore the research hotspots, gaps, and identify future research needs. To the best of our knowledge, this is the first study to report the toxicity of BPA based on bibliometric analysis.

## 2. Materials and Methods

### 2.1. Literature Search

As a recommended database and used widely in the bibliometric and citation network analysis [12,22,23,25], “Web of Science Core Collection” was chosen to conduct the literature search. Topic search, which searches the title, abstract, author keywords, and keywords plus in Web of Science, can obtain more reliable datasets than keyword search results when exploring a rapidly growing field [26]. The search term was as follows: “BPA (Topic)” AND “toxicity (Topic)”. All articles before 2022 were considered. Since some articles lacked information, such as authors, keywords, publishing year, and address, all the records were exported to the format as “Excel” with “full record” and then screened manually. Then all retrieved documents were exported by setting the export content to “Full record and cited references” and the export format to “Plain Text File”.

### 2.2. Research Tool and Data Process

All the refined documents were uploaded into the “VOSviewer” (version: 1.6.18) software, which performed data statistics and mapping on the literature information, including countries, institutions, journals, and keywords. When analyzing “co-authorship”, the unit of analysis was “organizations” and “countries”, respectively; as “co-occurrence” was selected in the analysis type, “All keywords” was the unit of analysis when exploring “co-occurrence”; for “citation” analysis, “documents”, “sources”, and “countries” in the analysis unit column were selected.

There are three kinds of visual maps in VOSviewer: network visualization, overlay visualization, and density visualization. In the network visualization, items (such as authors, countries, journals, and documents) are displayed with labels and colored circles connected with lines. The higher the weight of an item (generally refers to the frequency of occurrence), the larger the label and the circle of the item. The colors represent different clusters according to the relatedness of the items. In the map, the distance between two items also represents their correlation.

The overlay visualization looks similar to network visualization but with different colors. The color here represents the time when the items occurred, which is indicated by the timeline in the lower right corner.

In the density visualization, the label and the color representation of items are similar to those in network visualization but with bokeh background. In the article, the larger the number of items in the neighborhood of a point and the higher the frequency of the neighboring items, the brighter the color.

Before obtaining these maps, analysis outcomes are shown as tables to verify selected items. The table summarizes the item name, the number of documents, and citations and ranks them in total link strength.

## 3. Results

In total, 1644 documents were identified in the Web of Science database. After further manual screening, 32 publications were excluded for the lack of information, such as authors, publication year, and address (organization affiliation) (Figure 1). One is a retracted publication, which was removed from an academic journal. Finally, 1611 studies related to the toxicity of BPA were included in the bibliometric analysis. There are seven types of documents included according to the classification of the Web of Science database: Article; Article, data paper; Article, proceedings paper; Editorial material; Letter; Proceedings paper; Review. The number of research articles (1446) accounted for 89.76%.

### 3.1. The Annual Trend of Literature

As shown in Figure 2, the annual publication quantity increased from 1 in 1991 to 234 articles in 2021. In the first decade from 1991 to 2000, the number of studies related to BPA toxicity was the least, with only 19 articles in this decade. In the second stage, 2001–2009, the annual publications were between 10 and 30, which was relatively average. Between 2010 and 2015, the annual publication number gradually increased from 50 to 74. As the first jurisdiction to question BPA in the world, Canada declared BPA “toxic” in 2010, which may have caused widespread concern in industries as well as in the academic community and set off a wave of toxicological research. The Consortium Linking Academic and Regulatory Insights on BPA Toxicity (CLARITY-BPA), developed by the Food and Drug Administration, the National Toxicology Program, and the National Institute of Environmental Health Sciences in the U.S. in 2010, also played a significant role in BPA research. The release of research results from CLARITY-BPA in 2018 may partly account for the substantial growth in research documents between 2018 and 2021. As more and more countries and organizations have issued regulations and set health guidance values to constrain the use of BPA in recent years [27,28,29], academia raised the research boom on BPA.

### 3.2. Analysis of Research Countries

In total, seventy-nine countries recorded published papers on “BPA toxicity” between 1991 and 2021. The country distribution map of “BPA toxicity” literature is displayed in Figure 3. Results showed that China published the most documents on the topic, with a total of 486 papers, followed by the USA (313), South Korea (111), Japan (94), India (75), Spain (73), Italy (67), France (64), Germany (62), and Canada (59). The figure also reveals the cooperation among countries. The connection of studies between China and Ghana, and Ireland and Tanzania was relatively close. The USA conducted more cooperative research with Argentina and Ukraine, while South Korea had extensive cooperation with India, Japan, Kuwait, Malaysia, Oman, Portugal, the United Arab Emirates, and Vietnam.

Figure 4 shows the network map between countries based on the number of times they cited each other. The result revealed that researchers from the USA, Japan, Germany, England, Argentina, Czech Republic, Sweden, Finland, and Singapore started to explore BPA toxicity earlier. South Korea, Spain, France, Italy, Canada, Portugal, Switzerland, Greece, Belgium, and other countries focused on BPA toxicity after 2014. As an emerging country in the field of BPA toxicity, China witnessed an explosive research boom after 2016, with the most papers on BPA toxicity published. Libya, Ghana, Slovakia, Cameroon, and Morocco started toxicological research on BPA in recent years and cited the publication of the above countries.

In Table 1, information on each country’s number of documents and citations according to the total link strength was presented, showing China, the USA, and Japan as the top three countries with the largest total number of citations. China had the strongest connection (total number of citations). Still, the number of average citations per publication was relatively low compared to that of other countries, e.g., Germany, Japan, Canada, and the USA. One of the possible reasons may be the late start of Chinese research in this area compared to other developed countries.

### 3.3. Analysis of Research Institutions

From the perspective of institutions in co-authorship analysis, a total of 1815 was found in terms of affiliated institutions with published papers related to BPA toxicity. As shown in Table 2, the Chinese Academy of Sciences ranked first in the total link strength based on the number of co-authored documents. In the top 10 institutions, 6 of them were from the USA. Brunel University (UK) published only four papers, but the average number of citations of a single document (148.5) was the highest. These four articles were all co-authored by institutions from different countries, which may indicate that the quality of articles published by multi-national collaboration is higher and the influence is wider. Among the four documents, the most cited article was a review, assessing the effects of environmental concentration of BPA on wildlife, which was conducted by the USA, Japan, and England [30]. On the contrary, the Chinese Academy of Sciences and the University of Chinese Academy of Sciences published the highest number of papers, but their average citations per article were the lowest.

### 3.4. Analysis of Highly Cited Journals and Documents

The 1611 articles related to “BPA toxicity” were published in 477 journals. Table 3 shows the top 10 SCI journals cited in the field based on literature citations, in which the minimum document of each journal was set as “5” and 62 journals reached the threshold. The top 10 journals published 463 papers on “BPA toxicity” research, accounting for 28.74% of all the articles. The journal “*Chemosphere*” published the most studies on the topic, with a total of 3329 citations. The overlay visualization map of high-cited SCI journals in the field of “BPA toxicity” research is shown in Figure 5, “*Archives of Toxicology*”, “*Environmental Health Perspectives*”, and “*Mutation research genetic toxicology and environmental mutagenesis*” were the highly cited journals. In contrast, many cited documents in recent years were from “*Science of the Total Environment*”, “*Environmental Science and Pollution Research*”, and “*Environmental pollution*”.

Table 4 presents a summary of the top 10 documents based on total citations of all the 1611 papers analyzed. The two documents with the highest total link strength were both reviews [3,31], which reviewed the source, toxicity, biomonitoring, and human exposure of BPA and its analogs. Tyl et al. [32,33] studied the reproductive toxicity of dietary BPA in rats and mice and investigated the sensitivity of the two species to BPA. These two articles have received extensive attention. The paper with the most citations (831) was published by Vom Saal et al. [34] in 2005. This commentary clarified the definition of “Low Dose” and the status of literature on low-dose effects of BPA and explored factors accounting for the absence of significant adverse effects in low-dose BPA experiments. Their proposal of a new risk assessment for BPA and discussion on the challenge of “low dose” toxicity gained a lot of attention. Among the other documents listed in Table 4, two of them [30,35] are reviews outlining the harmful effects of BPA and other bisphenols. One of the other top-cited documents measured the concentrations of bisphenol analogs (BPs) in surface water from Japan, China, Korea, and India [36], and another one studied the toxicity of BPs to Zebrafish Embryo-Larvae [37]. The 10th top-cited study reported the concentration of bisphenol S in the urine of humans [38]. These high-cited documents indicate that reproductive toxicity, low-dose effects, human exposure, and toxicity and exposure to bisphenol analogs of BPA received extensive attention.

### 3.5. Research on Co-Occurring Keywords

In total, 7119 keywords were identified by co-occurring keywords analysis in VOSviewer. By setting “25” as the minimum number of keyword occurrences, 102 met the threshold. The time trend of “BPA toxicity” research keywords is shown in Figure 6. In the early years, researchers were more focused on “animal experiments” and “estrogenic activity” of BPA, which can be inferred from the co-occurrence keywords at that period, such as rats, mice, chemicals, reproduction, and estrogenic activity. However, in recent years, oxidative stress, apoptosis, damage, mitochondria, mechanism, neurotoxicity, bisphenol S, bisphenol F, analogs, alternatives, and human health have become new hotspots, indicating that researchers are exploring the toxic mechanism of BPA on the cellular level and focusing on its analogs. Table 5 lists the top 10 co-occurrence keywords in the number of total link strengths. In addition to the search terms (“BPA” and “toxicity”), the most frequent keywords were “exposure”, “oxidative stress”, “in-vitro”, “expression”, “reproductive toxicity”, and “endocrine-disrupting chemicals”. This showed that exploring the endocrine-disrupting mechanism, particularly reproductive toxicity of BPA through in-vitro experiments, attracted a considerable amount of research in the last three decades.

The co-occurrence keywords cluster density map is shown in Figure 7. Five major keyword clusters were obtained in VOSviewer according to their correlation: (1) the effect of BPA on reproduction and development with in-vivo experiments, which can be inferred from the keywords, such as reproductive toxicity, developmental toxicity, neonatal exposure, perinatal exposure, utero exposure, rat, mice, etc.; (2) the kinetics and physicochemical properties of BPA, which can be inferred from the keywords, such as adsorption, removal, kinetics, oxidation, degradation, liquid-chromatography, etc.; (3) the exposure of BPA and molecular mechanism of its toxicity, such as apoptosis, DNA damage, oxidative stress, cells proliferation, etc.; (4) mainly the alternatives of BPA, especially BPS and BPF; and (5) the combined impact of BPA and phthalates (one of the unbound chemicals in plastics) [39] /triclosan (an endocrine-disrupting compound) [40].

Our bibliometric analysis focused on the toxicology of BPA, which depicted the profile, hotspots, and research gaps regarding BPA toxicity. Yuan et al. [41] evaluated the hotspots and further trends in BPA and thyroid hormones using a bibliometric method. The results about the countries with the most papers/influential institution, authors, and journals are different from our study’s results. They emphasized the impact of BPA on thyroid hormones, especially in pregnant women and children. They showed that the implications for future research directions on toxicity of BPA are effect specific.

There are some limitations of this study. First, the analysis is based on publications in the Web of Science Core Collection, which is a comprehensive and reliable database for bibliometric analysis. Though the database is thorough enough, we still might have missed a few articles on this topic. Second, our research focused on the overall toxicology research of BPA, which may not be specifically indicative of one particular BPA toxic effect.

## 4. Conclusions

This study analyzed 1611 research articles about BPA’s toxicity through bibliometric methods. The following is a summary of our conclusions and recommendations.

(1)Research on the toxicity of BPA has advanced rapidly in the past three decades, and the annual publication number is still growing. This indicates that BPA has received more and more attention, but its toxic mechanism is still being explored.(2)China and the United States are in the leading research position, with a large number of articles and citations in this field. Among institutions, the Chinese Academy of Sciences (China) and the Dow chemical company (the USA) play an important role.(3)“*Chemosphere*” has published the most documents, which shows great interest in BPA toxicology. It may be a good source for researchers to collect and disseminate papers in the field.(4)Besides the search items (“BPA” and “Toxicity”), “Exposure”, “Oxidative stress”, and “In-vitro” are three main keywords, indicating that they are the research hotspots. The past 30 years have witnessed a shift in research focus from animal experiments and the estrogenic activity of BPA to the toxic mechanism of BPA and its analogs at the cellular level. The research on BPA is more and more extensive and in depth.(5)The current trends may indicate research gaps in our understanding on the toxicity mechanisms of these compounds and shifting levels of exposure in humans. Therefore, future research should probably focus on molecular biology and exposure assessment in humans from BPA and its substitutes.

In summary, this study analyzed the research status, identified the global trends and hotspots in BPA toxicology, and, thus, provided insights into future research direction.

## Figures and Tables

**Figure 1 ijerph-19-07886-f001:**
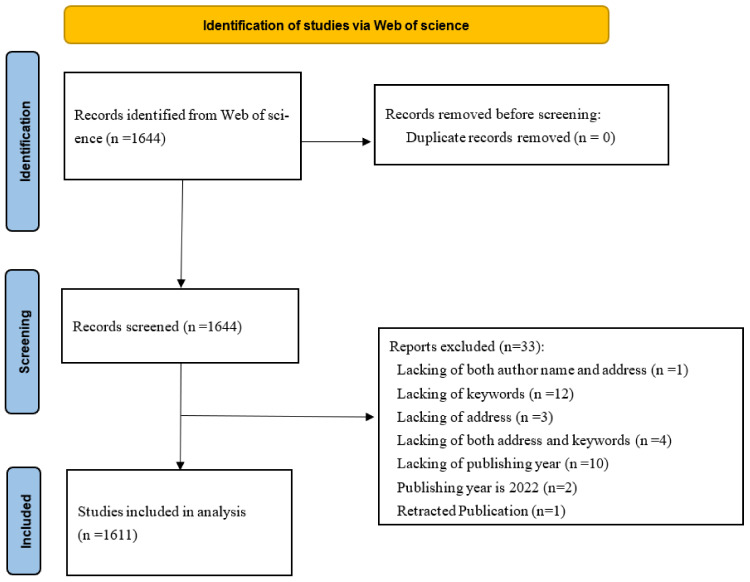
The flow chart of searching and selecting literature.

**Figure 2 ijerph-19-07886-f002:**
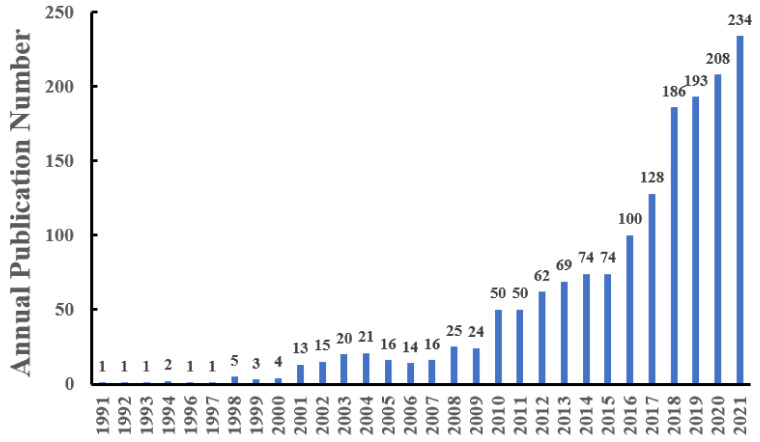
Annual publication number between 1991 and 2021.

**Figure 3 ijerph-19-07886-f003:**
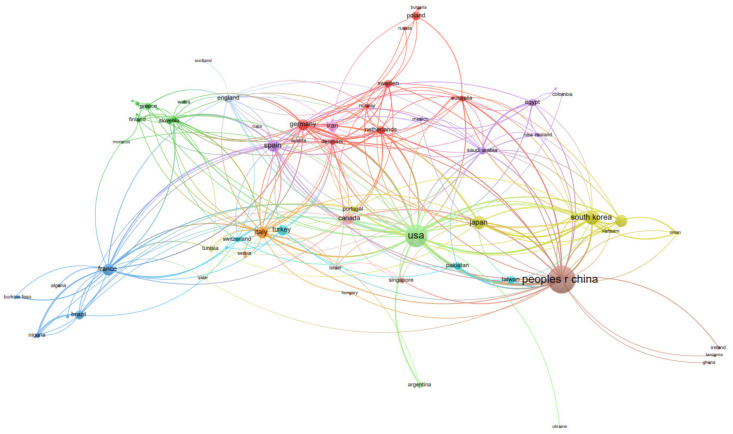
Country distribution of “BPA toxicity” literature according to co-authorship analysis.

**Figure 4 ijerph-19-07886-f004:**
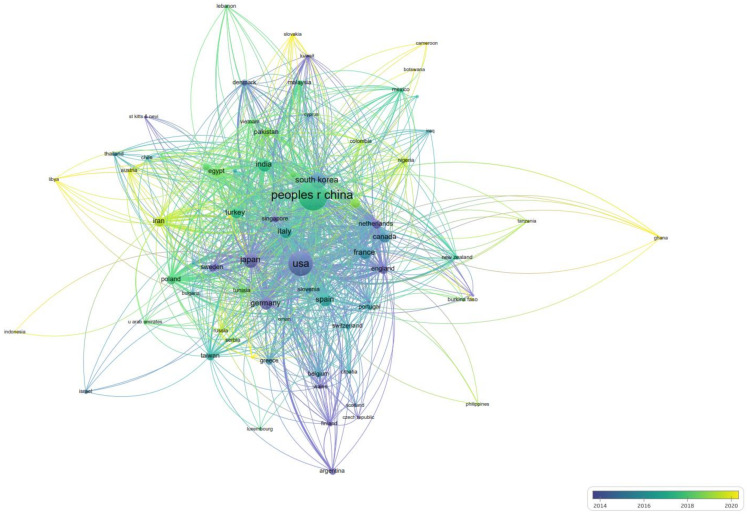
Country distribution of “BPA toxicity” literature according to citation times.

**Figure 5 ijerph-19-07886-f005:**
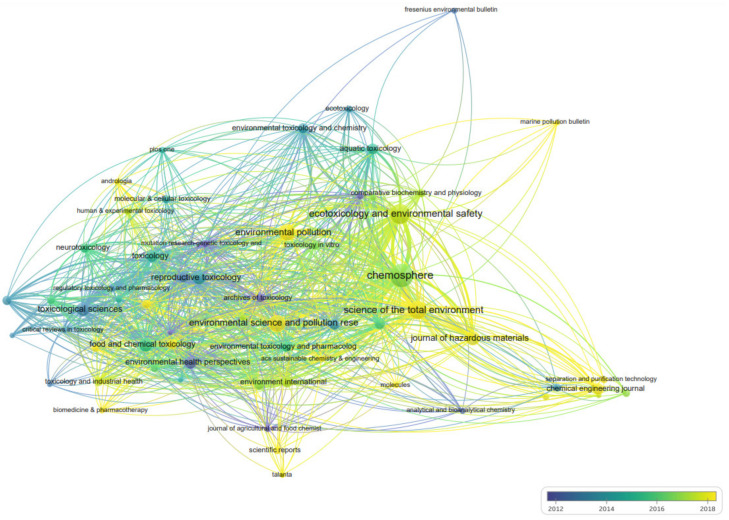
Journal distribution of “BPA toxicity” literature according to citation years.

**Figure 6 ijerph-19-07886-f006:**
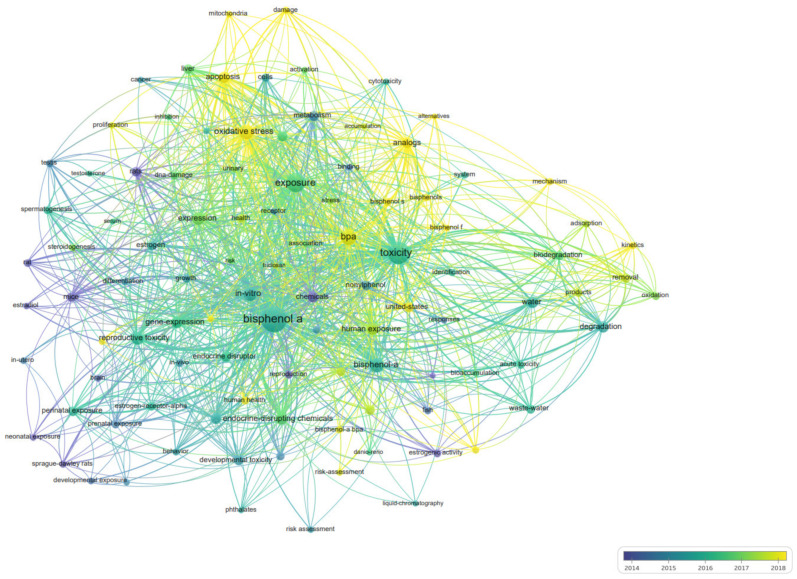
Overlay visualization of co-occurrence keywords for “BPA toxicity”.

**Figure 7 ijerph-19-07886-f007:**
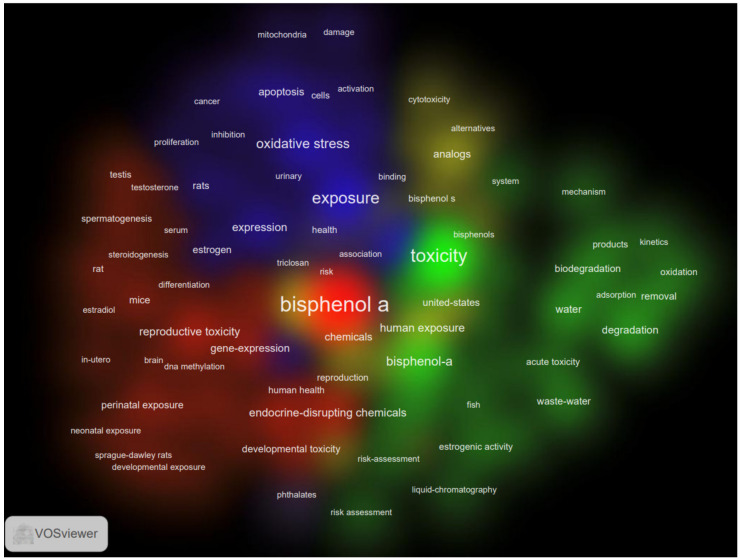
Cluster density of “BPA toxicity” co-occurrence keywords.

**Table 1 ijerph-19-07886-t001:** The number of documents and citations of top 10 countries.

Country	Documents	Citations	Average Number of Citations per Document	Total Link Strength ^1^
China	486	11,541	23.75	3998
USA	313	15,283	48.83	3857
Japan	94	5897	62.73	1622
South Korea	111	3043	27.41	1184
Germany	62	4557	73.50	1138
India	75	1762	23.49	938
Canada	59	2884	48.88	894
Spain	73	2445	33.49	849
Italy	67	2019	30.13	654
Poland	35	1256	35.89	648

^1^ The total link strength indicates the total strength of the citation links of a given country with other countries.

**Table 2 ijerph-19-07886-t002:** The number of documents and citations of top 10 organizations (ranked according to their link strength).

Organization	Documents	Citations	Average Number of Citations per Document	Total Link Strength ^1^
Chinese Academy of Sciences (China)	69	2032	29.45	121
The Dow chemical company (USA)	13	1036	79.69	68
Brunel University (UK)	4	594	148.50	57
University of Granada (Spain)	11	527	47.91	57
University of Cincinnati (USA)	17	1123	66.06	56
University of Chinese Academy of Sciences (China)	25	407	16.28	55
American Chemistry Council (USA)	9	544	60.44	54
University of Massachusetts (USA)	14	924	66.00	53
Washington State University (USA)	7	421	60.14	51
University of Missouri (USA)	10	1232	123.20	50

^1^ The total link strength indicates the total strength of the co-authorship links of a given institution with other institutions.

**Table 3 ijerph-19-07886-t003:** The number of documents and citations of top 10 Journals.

Journal	Documents	Citations	Total Link Strength ^1^	Impact Factor (2021)
Chemosphere	97	3329	938	7.086
Ecotoxicology and Environmental Safety	70	1967	770	6.291
Toxicological Sciences	32	1775	679	4.849
Reproductive Toxicology	32	1699	596	3.143
Science of the Total Environment	61	1468	514	7.963
Environmental Pollution	46	931	489	8.071
Environmental Science & Technology	20	1898	466	9.028
Environmental Science and Pollution Research	50	790	417	4.223
Food and Chemical Toxicology	31	996	406	6.023
Toxicology	24	1067	403	4.221

^1^ The total link strength indicates the total strength of the citation links of a given journal with other journals.

**Table 4 ijerph-19-07886-t004:** Summary of titles and citations of top 10 documents.

Document	Title	Citation	Total Link Strength ^1^
Chen et al. (2016) ([3])	Bisphenol Analogues Other Than BPA: Environmental Occurrence, Human Exposure, and Toxicity—A Review	598	164
Michałowicz J. (2014) ([31])	Bisphenol A—Sources, toxicity and biotransformation	475	149
Tyl et al. (2002) ([32])	Three-Generation Reproductive Toxicity Study of Dietary Bisphenol A in CD Sprague-Dawley Rats	351	148
Tyl et al. (2008) ([33])	Two-Generation Reproductive Toxicity Study of Dietary Bisphenol A in CD-1 (Swiss) Mice	234	121
Vom Saal et al. (2005) ([34])	An Extensive New Literature Concerning Low-Dose Effects of Bisphenol A Shows the Need for a New Risk Assessment	831	96
Crain et al. (2007) ([30])	An ecological assessment of bisphenol-A: Evidence from comparative biology	378	94
Yamazaki et al. (2015) ([36])	Bisphenol A and other bisphenol analogues including BPS and BPF in surface water samples from Japan, China, Korea and India	259	84
Moreman et al. (2017) ([37])	Acute Toxicity, Teratogenic, and Estrogenic Effects of Bisphenol A and Its Alternative Replacements Bisphenol S, Bisphenol F, and Bisphenol AF in Zebrafish Embryo-Larvae	189	76
Chen et al. (2002) ([35])	Acute toxicity, mutagenicity, and estrogenicity of bisphenol-A and other bisphenols	352	74
Liao et al. (2002) ([38])	Bisphenol S in Urine from the United States and Seven Asian Countries: Occurrence and Human Exposures	403	74

^1^ The total link strength indicates the total strength of the citation links of a given document with other documents.

**Table 5 ijerph-19-07886-t005:** The top 10 co-occurrence keywords for “BPA toxicity”.

Keyword	Occurrence	Total Link Strength ^1^
Bisphenol a	730	3189
Toxicity	530	2308
Exposure	369	1779
Oxidative stress	218	1100
BPA	243	1093
In-vitro	174	833
Bisphenol-a	198	746
Expression	147	729
Reproductive toxicity	135	639
Endocrine-disrupting chemicals	132	622

^1^ The total link strength indicates the total strength of the co-occurrence links of a given keyword with other keywords.

## Data Availability

Not applicable.

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
