# Peer review of "Bibliometric Analysis of the Toxicity of Bisphenol A"

_ijerph, 2022, doi:10.3390/ijerph19137886_

Round 1

Reviewer 1 Report

This review paper is well prepared and has a significant contribution to the field, I recommend accept after minor revision.

1. line 49-50, merge these two paragraphs

2. line 56-63, merge these two paragraphs

3. provide sharper and higher quality figures

4. checking the English language by a native speaker

Reviewer 2 Report

I have read with interest the present paper, which was aimed to investigate the WOS database to identify the general trends in BPA toxicology to highlight the direction of current (and future) research on the subject. In general the paper is clear, well written, and paragraphs are organized with clarity. In my opinion, this paper successfully centers its descriptive goal, but somehow suffers from a rather plain interpretation of the data presented. If possible, I would recommend to expand the conclusions with a few sentences discussing the main findings.

I also suggest the authors to address a few other points:

Lines 39-55 - I feel this part is a bit beyond the scope of the paper. The first lines describe sufficiently the bibliometric analyses, perhaps the second part could be summarized?

Line 70 - wouldn't the inclusion of just two keywords ("BPA" AND "toxicity") limit too much the search? Why a broader database (for example Medline) was not used? Please comment.

Line 106 - How many retracted articles were included?

Reviewer 3 Report

Dear editor and authors

This is a review paper for Bisphenol A toxicity with VOSviewer as a tool for bibliometric analysis from 1991 to 2021.

1. Major issue

As for the past 30 years, Hong Kong and Taiwan are much separated from China. 

The result of " Taiwan and bisphenol a" and " Hong Kong and bisphenol a"

contribute for 206 and 192 results, respectively from 1986 to 2022 on pubmed.

( For cmparison, "South Korea bisphenol a" revealed 464 results within the similar time interval) 

These results implied that Taiwan and Hong kong may share an important part in the research (the 3 circles in Figure 1, which all labelled as "people r china" are likely to be Taiwan, Hong Kong, China, from left to right) A clear-described information may provide researchers more detail where and how to design their study in the furture.  Please correct it accordingly.

Another question for the countries distribution in figure 3, is that" where is the country of Japan?" 

Japan has many famous make-up companies. And The result of " Japan and bisphenol a" contribute for 1295 results from 1987 to 2022 on pubmed. 

It is highly unlikely that Japan is not showing the countries who release studies of  "bisphenol a". ( Besides, Japan and Taiwan appeared in the Figure 4)

2. Minor 

a. 

There is a similar study of "Bisphenol A and thyroid hormones -Bibliometric analysis of scientific publications " in 2020, which is using exactly the same tool "VOSviewer" for analysis. So to our readers, this is a "metoo" study using different keywords. 

Medicine (Baltimore). 2020 Nov 6; 99(45): e23067.

Published online 2020 Nov 6. doi: 10.1097/MD.0000000000023067

What is the new information provided uniquely by this study? And why the authors decide to do a similar study after background research for this topic? It needs more explanation.

b.

The Fig 6 and Fig 7 looks much alike, what is the new information provided by Fig 7? If not much, Fig 7 may be deleted or merged with Fig 6.

c.

English polishing may be needed after revision.

Round 2

Reviewer 2 Report

I feel that most of the issues I raised have been satisfactorily answered. I only believe the authors should only clarify the answer to the last point I raised. In other words, the revised manuscript on line 107 lists retracted publication as an article type included in the analysis. I believe it is necessary to specify whether and how many of these have been included.

Author Response

Thanks for your professional comment. We listed retracted publication as an article type according to the search result of the Web of Science database. But we focused on the research articles and ignored this type of article. As we know, once a published paper is removed from the journal, it means it is unreliable data. So we think it should not be included in the analysis. There is one retracted publication in our search result. We excluded this article and reanalyzed the data. The related results were adjusted accordingly. You can check it in the revised manuscript. All revisions made to the manuscript were marked up using the “Track Changes” function.

Reviewer 3 Report

agree with revision 

Author Response

Thanks for your comments. We checked the article and made some modifications. You can check it in the revised manuscript. All revisions made to the manuscript were marked up using the "Track Changes" function.